# Positive Energy and Non-SUSY Flows in ISO(7) Gauged Supergravity

**Giuseppe Dibitetto** [1,2]

1  Dipartimento di Fisica e Astronomia, Università di Padova, Via Marzolo 8, 35131 Padova, Italy; giuseppe.dibitetto@pd.infn.it
2  INFN, Sezione di Padova, Via Marzolo 8, 35131 Padova, Italy

**Abstract:** We consider maximal gauged supergravity in 4D with the ISO(7) gauge group, which arises from a consistent truncation of massive IIA supergravity on a six-sphere. Within its $G_2$-invariant sector, the theory is known to possess a supersymmetric AdS extremum, as well as two non-supersymmetric ones. In this context, we provide a first-order formulation of the theory by making use of the Hamilton–Jacobi (HJ) formalism. This allows us to derive a positive energy theorem for both non-supersymmetric extrema. Subsequently, we also find novel non-supersymmetric domain walls (DWs) interpolating between the supersymmetric extremum and each of the other two. Finally, we discuss a perturbative HJ technique that may be used in order to solve for curved DW geometries.

**Keywords:** quantum gravity; anti-de Sitter space; string compactifications

## 1. Introduction

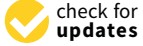



One of the most controversial aspects of string phenomenology is the possible existence of a huge *landscape* of vacua of the theory, covering a wide range of lower-dimensional features. In the original argument of [1], the string landscape was estimated to contain $\mathcal{O}(10^{500})$ different vacua, while a more recent F-theory analysis extends it even further to $\mathcal{O}(10^{272000})$ [2]. However, these estimations are usually thought of as very rough, since they try to count all distinct sets of topological data of the compact manifold, together with all possible internal flux configurations, without though really taking the dynamics into account.

On the other hand, regardless of what the exact order of magnitude may be, the concept itself of such a wide landscape seems to spoil the predictivity of string theory and, moreover, appears to be in tension with the idea of a theory of everything with no free parameters. By adopting a proper *top-down* approach, one might say that the only lower-dimensional constructions that have all rights to be in the landscape are those that can be directly obtained by a string compactification within the right perturbative corner, i.e., where quantum loop, higher derivative, and instanton corrections are all under control.

This way, one would naturally exclude any other kind of lower-dimensional EFT that seems perfectly consistent from a low energy viewpoint, and yet, it has no known string theory origin. All such constructions should then be relegated to the so-called *swampland*. Nevertheless, since there might be constructions that do admit a stringy embedding that is simply very difficult to find or just not yet fully understood, it may be worthwhile to look for a set of purely *bottom-up* criteria that one can use to distinguish the landscape from the swampland. This was exactly the philosophy proposed by the authors of [3] when they formulated the *weak gravity conjecture* (WGC) as the first swampland criterion. This criterion states that any consistent EFT must retain gravity as the weakest force in the game, at any energy scale. As a consequence, within an EFT with gauge interactions controlled by

a coupling $g$, the existence of microscopic particles (with mass smaller than the charge in Planck units) is needed in order to obstruct taking the ungauged limit of the theory $g \to 0$.

In more recent years, a whole *string swampland program* [4] has been developing with the aim of defining an entire set of swampland criteria based on quantum gravity principles, which should help us select good stringy EFT's from the rest. Of course, to start with, these criteria are formulated as conjectures, which are supported by stringy insights, and only checked explicitly in some specific string theory setups. This way, we came to conceive of a web of different swampland criteria, which become more and more refined and better understood as time goes by, and interesting relations among all of them arise. For a complete review of the topic, we address the reader to [5] and the references therein.

In this paper, we want to focus our attention on the so-called *AdS swampland conjecture* proposed by [6], according to which all non-supersymmetric AdS vacua must be ultimately unstable in string theory. The intuition behind this relies on a stronger version of the WGC applied to membranes (see also [7] for a similar logic). According to this, if the vacuum is supported by flux (which is generically the case), it will be unstable against spontaneous nucleation of microscopic membranes (tension smaller than the charge), which will eventually discharge the flux and, hence, destroy the associated AdS vacuum. Note that such a process is intrinsically non-perturbative, and it does not affect supersymmetric vacua, since the corresponding spectrum *only* has BPS membranes (tension equal to the charge).

Though the intuitive picture is very clear, there are still very few explicit examples of such stringy constructions within a controlled setup (see, e.g., the one in [8], where spherical D3 brane nucleation is found to destroy the non-supersymmetric and, yet, tachyon-free orbifolds of $AdS_5 \times S^5$). In this paper, we study non-supersymmetric $AdS_4 \times S^6$ vacua of massive IIA supergravity. Such solutions are captured by a 4D gauged supergravity theory with gauge group $ISO(7)$ [9] and, in particular, its $G_2$-invariant subsector. Remarkably, its $G_2$-preserving non-supersymmetric vacuum has already passed all perturbative checks, including the computation of the full KK spectrum [10], as well as brane jet instabilities of different sorts [11]. One really needs a non-perturbative analysis to explicitly check the validity of the AdS swampland conjecture in this case.

In this paper, we try to take a complementary approach and check for the possible existence of a dynamical protection mechanism for the aforementioned non-supersymmetric vacuum, by trying to derive a positive energy theorem, analogous to the one protecting supersymmetric AdS [12]. We will study the problem within a 4D $\mathcal{N} = 1$ supergravity model capturing both the supersymmetric extremum, as well as the non-supersymmetric one. The technical tool we will use is the Hamilton–Jacobi (HJ) formalism, which will allow us to derive the aforementioned positive energy theorem. It is worth stressing that this has *no direct implications* concerning full non-perturbative stability of the above vacuum. The reason for this is that what we conventionally refer to as the 4D reduced model is not a good EFT in a Wilsonian sense; it simply captures a self-contained sector of the theory in a mathematical sense. Therefore, the existence of a positive energy theorem will simply tell us that potential non-perturbative instabilities, if any, must be looked for elsewhere. Indeed, while the vacuum is found to be non-perturbatively stable by virtue of the positive energy theorem within this sector, a non-perturbative instability completely analogous to the one in [8] is found in the companion paper [13], where the role of D3 branes is taken there by D2 branes.

The paper is organized as follows. In Section 2, we introduce our setup, i.e., a particular 4D minimal supergravity model admitting SUSY, as well as non-SUSY AdS extrema. In Section 3, we discuss the central role of first-order flows when studying bubble nucleation and positive energy theorems. In Section 4, we focus on radial flows featuring a codimensional one flat slicing of the metric and introduce their HJ formulation. In Section 5, we review a general positive energy theorem and give a constructive proof thereof in our reduced 4D model. This will imply a dynamical protection from decay for the non-SUSY

vacua within our model. In Section 6, we briefly discuss radial flows involving a curved slicing of the 4D metric. Within this context, we present a viable algorithm that allows one to perturbatively integrate the HJ problem at any desired order. Finally, we collect in Appendix A some details concerning the massive IIA supergravity origin of the 4D reduced model studied here.

## 2. The Setup: $G_2$-Invariant ISO(7) Gauged Maximal Supergravity

ISO(7) gauged $\mathcal{N} = 8$ supergravity [14] is a consistent gauging of maximal supergravity in $D = 4$ belonging to the $CSO(p, q, 8 - p - q)$ class, i.e., all gauge groups obtained from SO(8) [15] by performing analytic continuations, as well as Inönü–Wigner contractions [16]. In these gaugings, the gauge group is embedded in the full $E_{7(7)}$ global symmetry of the ungauged theory through its maximal $SL(8, \mathbb{R})$ subgroup. The corresponding embedding tensor of the theory lies within the $\mathbf{36} \oplus \overline{\mathbf{36}}$ of $SL(8, \mathbb{R})$:

$$
\begin{aligned}
E_{7(7)} &\supset SL(8, \mathbb{R}), \\
\mathbf{912} &\to \underbrace{\mathbf{36}}_{Q^{(MN)}} \oplus \underbrace{\overline{\mathbf{36}}}_{P_{(MN)}} \oplus \mathbf{420} \oplus \overline{\mathbf{420}},
\end{aligned}
\tag{1}
$$

All consistent $CSO(p, q, 8 - p - q)$ gaugings are then parametrized by the two symmetric $8 \times 8$ matrices $Q$ and $P$ solving the following quadratic constraint:

$$
\text{Tr}(QP) = Q^{MN} P_{MN} \overset{!}{=} 0.
\tag{2}
$$

The standard ISO(7) theory as arising from the reduction of *massless* IIA supergravity on a six-sphere [17] corresponds to the choice:

$$
Q = \left( \begin{array}{c|c} 0 & \\ \hline & q\,\mathbb{I}_7 \end{array} \right) \quad \text{and} \quad P = \mathbb{O}_8.
\tag{3}
$$

However, one may turn on an extra parameter within this gauging, which essentially also takes a possible symplectic deformation [18] into account. This gives rise to

$$
Q = \left( \begin{array}{c|c} 0 & \\ \hline & q\,\mathbb{I}_7 \end{array} \right) \quad \text{and} \quad P = \left( \begin{array}{c|c} p & \\ \hline & \mathbb{O}_7 \end{array} \right),
\tag{4}
$$

which of course solves the quadratic constraint (2), as well as the standard ISO(7) theory did. In type IIA language, the new $p$ parameter controlling the symplectic deformation turns out to correspond to Romans' mass, and hence, the theory can be obtained as a consistent truncation of *massive* IIA supergravity on a six-sphere [19]. Some details concerning the truncation to its $G_2$-invariant sector are collected in Appendix A. As we will see later on, contrary to its massless counterpart, this theory contains non-trivial AdS$_4$ vacua.

The 32 real supercharges of the theory are organized into the fundamental representation of the R-symmetry group $SU(8)_R$. When restricting to the $G_2$-invariant sector, these decompose as

$$
\begin{aligned}
SU(8)_R &\supset SO(8) \supset SO(7)_+ \supset G_2 \\
\mathbf{8} &\to \mathbf{8}_v \to \mathbf{8} \to \mathbf{1} \oplus \mathbf{7}
\end{aligned}
\tag{5}
$$

Hence, the truncated theory enjoys $\mathcal{N} = 1$ supersymmetry. The 70 physical scalars of maximal supergravity decompose as

$$
\mathbf{70} \to \mathbf{35}_v \oplus \mathbf{35}_s \to \mathbf{1} \oplus \mathbf{7} \oplus \mathbf{27} \oplus \mathbf{35} \to \underbrace{\mathbf{1} \oplus \mathbf{1}}_{\text{one } \mathbb{C} \text{ scalar}} \oplus \mathbf{14} \oplus \mathbf{27} \oplus \mathbf{27},
\tag{6}
$$

where the $G_2$ singlet scalars are organized into one complex field named $S$. The corresponding embedding tensor deformations contain four real $G_2$ singlets in total, two coming from the $\mathcal{A}_1$ piece of the T-tensor and the remaining two coming from the $\mathcal{A}_2$ piece. However, out of these four parameters, only two remain independent when specifying to the ISO(7) gauging. In terms of minimal supergravity language, these two parameters generate the following superpotential deformation:

$$\mathcal{W} = 14g\,S^3 + 2m. \tag{7}$$

The Kähler potential determining the scalar kinetic coupling reads:

$$\mathcal{K}(S, \bar{S}) = -7\log(-i(S - \bar{S})). \tag{8}$$

The corresponding action is given by [19]

$$\mathcal{S}_{G_2}[g_{\mu\nu}, S, \bar{S}] = \int d^4x \sqrt{-g_4}\left(\frac{\mathcal{R}_4}{2\kappa_4^2} - K_{S\bar{S}}\,(\partial S)(\partial \bar{S}) - V(S, \bar{S})\right), \tag{9}$$

where $\kappa_4$ denotes the 4D gravitational coupling, $K_{S\bar{S}} \equiv \partial_S \partial_{\bar{S}} \mathcal{K}$, and the scalar potential $V$ is determined by the superpotential through[1]

$$V = e^{\mathcal{K}}\left(-3|\mathcal{W}|^2 + |D\mathcal{W}|^2\right), \tag{10}$$

where $D$ denotes the Kähler covariant derivative, e.g., $D_S\mathcal{W} \equiv \partial_S\mathcal{W} + \partial_S\mathcal{K}\,\mathcal{W}$. After fixing the specific scalar parametrization given by

$$S = \chi + ie^{-\varphi}, \tag{11}$$

the explicit expression of the scalar potential reads[2]

$$V = \frac{m^2}{8}\,e^{\varphi}\left(-35 - 21e^{2\varphi}\chi^2 + 63e^{4\varphi}\chi^4 + e^{6\varphi}(1 + 7\chi^3)^2\right). \tag{12}$$

This potential admits three different critical points, one of which is supersymmetric, while the remaining ones have spontaneously broken SUSY. With this choice of embedding tensor normalization, the three critical points are illustrated in Table 1, while in Figure 1, we show the level curves of the scalar potential around the three critical points.

**Table 1.** The three different AdS critical points of ISO(7) gauged supergravity preserving at least $G_2$ as residual symmetry. The squared masses of the $G_2$ singlet modes are normalized to the absolute value of the cosmological constant. In these units, the BF bound [20] lies at $m_{\mathrm{BF}}^2 = -\frac{3}{4}$.

| ID | SUSY | $G_{\mathrm{res.}}$ | $\chi$ | $e^{-\varphi}$ | $m_{G_2}^2$ |
|----|------|---------------------|--------|----------------|-------------|
| 1 | $\mathcal{N} = 0$ | SO(7) | 0 | $5^{-1/6}$ | $2; -\frac{2}{5}$ |
| 2 | $\mathcal{N} = 1$ | $G_2$ | $2^{-7/3}$ | $3^{1/2}5^{1/2}2^{-7/3}$ | $\frac{1}{3}\left(4 \pm \sqrt{6}\right)$ |
| 3 | $\mathcal{N} = 0$ | $G_2$ | $-2^{-4/3}$ | $3^{1/2}2^{-4/3}$ | $2; 2$ |

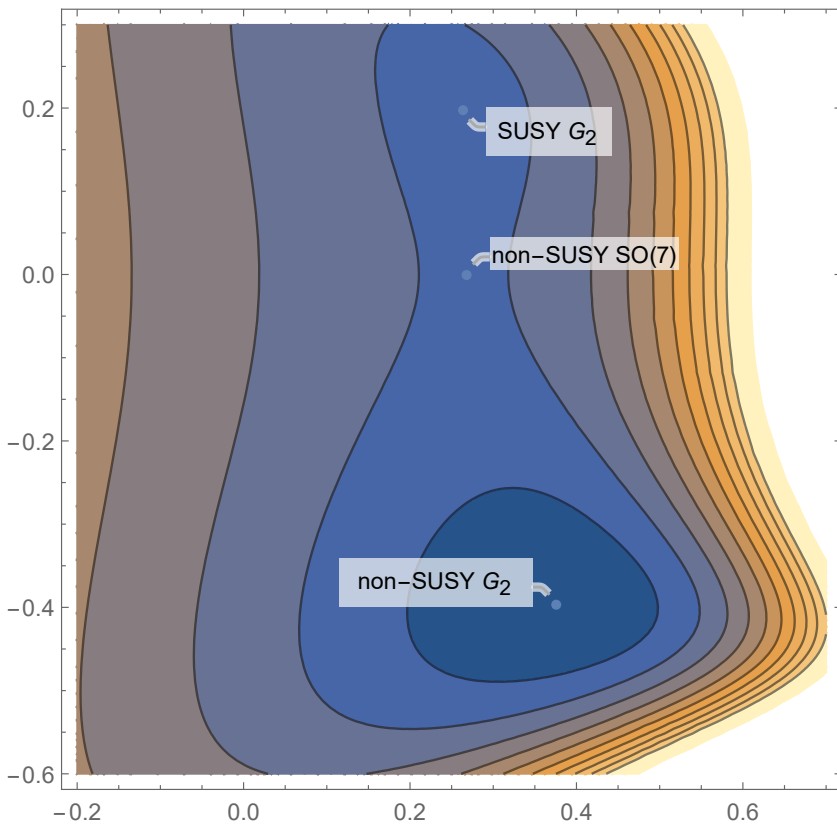

**Figure 1.** The level curves of the scalar potential for the $G_2$-invariant sector of ISO(7) gauged supergravity. The horizontal axis represents the $\varphi$ direction, while the vertical axis represents $\chi$. Both the SUSY and non-SUSY $G_2$ extrema are local minima, while the SO(7) critical point is actually a saddle.

In [11], the complete mass spectra for the entire set of scalar fields within maximal supergravity was computed, and while the SO(7) invariant critical point exhibits the presence of modes below the BF bound, the non-SUSY $G_2$ extremum remains completely *tachyon free*. In addition to this perturbative analysis, still in [11], a different stability analysis was performed, i.e., the one with respect to brane jets by making use of the probe approximation. As a result, the $G_2$ point was found to be also protected from this type of instability. Finally, in [10], the full KK spectrum was checked by using ExFT techniques, and even this calculation showed no signs of instabilities for this vacuum.

As a consequence, one might argue that this special example of the non-SUSY AdS vacuum of massive IIA supergravity poses a challenge to the common lore brought up by the swampland program, which generically expects instabilities to arise, both at a perturbative [21,22] and non-perturbative level [6], whenever SUSY is broken. In the following sections, we aim at studying possible dynamical protection mechanisms preventing non-perturbative instabilities, at least within controlled EFT setups such as the 4D $\mathcal{N} = 1$ description at hand.

We will indeed be able to derive a positive energy theorem for all vacua in this truncated theory that will exclude the occurrence of non-perturbative decays within this universal sector. As already stressed in the Introduction though, this will of course not imply the full stability of the corresponding vacua, but will only tell us that possible non-perturbative decay channels should be searched for within a more general setup. In analogy with [23], the technical tool used here for our scope will be the HJ formalism, which will allow us to translate the problem of the (non-)existence of certain radial flows into the (non-)existence of HJ generating functions with certain local and global properties.

### 3. First-Order Flows: Bubbles, DWs, and Positive Energy

The instabilities of a classical gravitational vacuum at a non-perturbative level have been studied since the 1980s by adopting a semiclassical approach. The relevant object in this context is the Euclidean path integral, which describes the nucleation probability of instantons. Within such a description, viable gravitational instantons are given by smooth bubble geometries that asymptote to a given false vacuum and have a finite Euclidean action. In more recent years, with the aid of string theory completions, also singular bubble solutions started to be considered in situations where the singularity can be resolved by a brane in string theory. In these cases, the corresponding non-perturbative decay process described by the associated geometry is the spontaneous nucleation of a charged spherical brane. Therefore, in a more general sense, non-perturbative instabilities of a given AdS vacuum may be seen as solutions to the reduced dynamics of gravity coupled to scalars, which describe a foliation of the lower-dimensional metric (4D, in our case) with an FRW slice in one dimension lower (3D, in our case):

$$ds_4^2 = e^{2A} ds_{\text{FRW}_3}^2 + e^{2B} dr^2, \tag{13}$$

where $A$, $B$, and the scalar fields are suitable functions of the radial coordinate $r$.

In order for this to actually represent a viable decay channel, one must of course make sure that the following requirements are met:

- The effective FRW cosmology on the bubble wall is of an expanding type;
- The asymptotic geometry is our (putative) AdS vacuum;
- The difference in Euclidean action $0 < S_{\text{E}}(\text{Bubble}) - S_{\text{E}}(\text{AdS}) < \infty$.

First of all, one can easily check that the dynamics of the system in the absence of coupling with extra matter implies the slicing to be *maximally symmetric*, hence AdS$_3$, Mkw$_3$, or dS$_3$, depending on the value of the curvature on the 3D slices. As a consequence, expanding bubbles will necessarily correspond to dS slicings of our AdS$_4$. Then, the specific type of instability will crucially depend on the global properties of the solution, and especially on its asymptotic behavior at the other boundary of the $r$ coordinate. If it asymptotes to a different AdS geometry, what we have is a CDL bubble [24] describing decay to a true vacuum through bubble nucleation. If it approaches a singular metric, we need to look at its UV description in terms of string/M-theory in order to understand the physics behind it. It could be a bubble of nothing (BoN) [25] if the singularity becomes resolved in higher dimensions into a smoothly shrinking Euclidean geometry, or conversely, it could correspond to a spherical brane nucleation process if there turns out to be a stringy source placed at the singularity, or again, it could instead correspond to a decompactification limit.

Secondly, the last condition given in terms of the variation of the Euclidean action may be viewed as a *zero-energy condition* for the bubble geometry. In other words, the Euclidean action is extremized at a finite value only if its asymptotically AdS geometry has zero energy with respect to empty AdS itself. For a CDL bubble in the thin wall limit, i.e., where the scalar fields are approximated by step functions and, hence, carry no kinetic energy along the radial flow, this zero-energy condition is understood as a bound on the tension of the (thin) bubble wall, which is usually referred to as the CDL bound. This condition was directly interpreted in [26] as a consequence of energy conservation in a bubble nucleation process. Later, in the classification of [27], bubble walls respecting the CDL bound were called *ultra-extremal* walls and were indeed found to lead to instabilities [28].

In this context, flat domain wall (DW) geometries represent a limit of the aforementioned bubble walls when the Euclidean action can only be extremized at an infinite value of the bubble radius, and this happens in turn when the CDL bound is exactly saturated. However, these do not correspond to real instabilities, but rather to *walls of marginal stability* separating two phases that would require infinite time to decay into each other. In some sense, one might already expect that the existence of interpolating solutions with flat slicing

(Mkw$_3$) could in itself exclude the existence of actual bubbles with the same initial and final points.

More precisely, it is within the same context of flows with flat slices that one can find the objects proving a positive energy theorem for a given AdS vacuum. By this, we mean that the existence of some special flat radial flows automatically implies that any bulk perturbations of the AdS asymptotic geometry will necessarily have positive energy, and hence no finite Euclidean action, and therefore no significant contribution to the path integral. In the next sections, we will first formulate HJ for flat flows and show its relation to fake supergravity, and subsequently use it to give a constructive proof of the positive energy theorem for the G$_2$-invariant vacua of ISO(7) gauged supergravity, as well as to find interpolating flat DWs.

## 4. Flat First-Order Flows: Separable HJ Treatment

Flat radial flows within the G$_2$-invariant sector of ISO(7) gauged supergravity may be found by exploiting the first-order formulation through the HJ formalism. These flows are solutions of the form

$$ds_4^2 = e^{2A}ds_{\text{Mkw}_3}^2 + e^{2B}dr^2, \tag{14}$$

where $A$ and $B$, as well as the scalar fields $\varphi$ and $\chi$ are functions of the radial coordinate $r$. It is worth mentioning that $B$ is actually pure gauge. However, as often in these cases, it may be useful to make use of this gauge-fixing freedom in order to better integrate the first-order flow.

When plugging *Ansatz* (14) into the action (9), one obtains the following 1D reduced Lagrangian[3]:

$$L_{\text{eff}} = e^{3A-B}\left(3(A')^2 - \frac{7}{4}(\varphi')^2 - \frac{7}{4}e^{2\varphi}(\chi')^2\right) - e^{3A+B}V(\varphi,\chi), \tag{15}$$

where $'$ denotes differentiation with respect to $r$, and $V$ is the scalar potential introduced in (12). It is worth stressing that evaluating the supergravity action on a specific *ansatz* as we do here does not lead to a consistent set of field equations in general. Nevertheless, in this particular case, it has been checked explicitly that the 1D Lagrangian (15) yields exactly the same set of field equations as those obtained from the 4D equations of motion evaluated on (14). Now, we introduce the conjugate momenta for our dynamical fields $(A, \varphi, \chi)$:

$$\begin{cases} \pi_A &= \dfrac{\partial L_{\text{eff}}}{\partial A'} = 6\,e^{3A-B}A', \\[2mm] \pi_\varphi &= \dfrac{\partial L_{\text{eff}}}{\partial \varphi'} = -\dfrac{7}{2}e^{3A-B}\varphi', \\[2mm] \pi_\chi &= \dfrac{\partial L_{\text{eff}}}{\partial \chi'} = -\dfrac{7}{2}e^{3A-B+2\varphi}\chi'. \end{cases} \tag{16}$$

By performing the usual Legendre transform, we obtain the following 1D Hamiltonian[4]

$$H_{\text{eff}} = e^{-3A}\left(\frac{\pi_A^2}{3} - \frac{4}{7}\pi_\varphi^2 - \frac{4}{7}e^{-2\varphi}\pi_\chi^2\right) + e^{3A}\,V(\varphi,\chi)\,. \tag{17}$$

According to the HJ formalism, the second-order dynamics associated with the Lagrangian (15) turns out be equivalent to the following first-order flow defined by a suitable generating function $F(A,\varphi,\chi)$:

$$\begin{cases} \pi_A = \dfrac{\partial F}{\partial A} \equiv F_A, \\[2mm] \pi_\varphi = \dfrac{\partial F}{\partial \varphi} \equiv F_\varphi, \\[2mm] \pi_\chi = \dfrac{\partial F}{\partial \chi} \equiv F_\chi, \end{cases} \tag{18}$$

when supplemented with the HJ constraint $H_{\mathrm{eff}}|_{\pi_I \to F_I} \overset{!}{=} 0$, which fixes the form of the generating function $F(A, \varphi, \chi)$. In this case, the HJ constraint takes the following explicit form:

$$e^{-3A}\left(\frac{F_A^2}{3} - \frac{4}{7}F_\varphi^2 - \frac{4}{7}e^{-2\varphi}F_\chi^2\right) + \underbrace{e^{3A}\, V(\varphi,\chi)}_{V_{\mathrm{eff}}(A,\varphi,\chi)} \overset{!}{=} 0. \tag{19}$$

Since the effective potential has a *factorized* dependence of the warp factor $A$, the corresponding generating functional may be found by casting a separable *ansatz* $F(A, \varphi, \chi) = e^{3A} f(\varphi, \chi)$, where the function $f$ of the scalar fields satisfies the following non-linear PDE:

$$-3f^2 + 2K^{ij}\partial_i f \partial_j f \overset{!}{=} V(\varphi, \chi)\,, \tag{20}$$

where $K^{ij} \equiv \mathrm{diag}\left(\frac{2}{7}, \frac{2}{7}e^{-2\varphi}\right)$ is nothing but the inverse kinetic metric. This identity manifestly shows the interpretation of these first-order flows as a realization of *fake supergravity* [29].

Indeed, one can show that

$$f_{\mathrm{SUSY}} \equiv e^{\mathcal{K}/2}\,|\mathcal{W}| = m\,\frac{e^{7\varphi/2}\sqrt{\left(14e^{-3\varphi} - 42e^{-\varphi}\chi^2\right)^2 + 4\left(7\chi\left(\chi^2 - 3e^{-2\varphi}\right) + 1\right)^2}}{8\sqrt{2}} \tag{21}$$

is a global solution to the PDE (20), corresponding to the actual superpotential of the theory. Thanks to the existence of the superpotential, one may construct a globally bounding function as $-3f_{\mathrm{SUSY}}^2$, which dynamically protects the SUSY vacuum from non-perturbative decay by virtue of the positive energy theorem for supersymmetry [12] (see also [30,31]).

## 5. Positive Energy Theorems for Non-SUSY Vacua

Now, because the PDE (20) characterizing the function $f$ is non-linear, $f_{\mathrm{SUSY}}$ may not be the only global solution. In [32], it was studied how solutions of the aforementioned PDE can imply dynamical protection from non-perturbative decay. The claim can be summarized into the following theorem [33].

**Theorem 1.** *If a scalar potential $V(\phi^i)$ in a given EFT can be written as*

$$V = -3f^2 + 2K^{ij}\partial_i f \partial_j f, \tag{22}$$

*for a suitable globally defined function $f$ of the scalars such that*

- *(i) $\partial_i f|_{\phi_0} = 0$,*
- *(ii) $V(\phi) \geq -3f(\phi)^2, \forall \phi \in \mathcal{M}_{scalar}$,*

*then other points in the scalar manifold $\mathcal{M}_{scalar}$ have higher energy than $\phi_0$ itself, whence $\phi_0$ is stable against non-perturbative decay within the given EFT.*

As remarked earlier, due to the non-linearity of the defining PDE, multiple global $f$s might exist. Moreover, since any critical point would represent a *singular* initial condition, multiple solutions are possible even at a local level. By following the analysis in [23], one may construct all of the local solutions of (20) by perturbatively expanding around each of

the critical points of the scalar potential. This way, we may look for explicit solutions of the form[5]

$$ f = \sum_{k=0}^{\infty} f^{(k)}(\phi_0) \frac{(\phi - \phi_0)^k}{k!} , \tag{23} $$

where the numerical coefficients $f^{(k)}(\phi_0)$ of the above expansion are in fact rank $k$ tensors. After fixing the zeroth- and first-order coefficients to be

$$ f^{(0)} = \sqrt{-\frac{V(\phi_0)}{3}} , \qquad \text{and} \qquad f^{(1)} = 0, \tag{24} $$

the second-order coefficients satisfy second-degree algebraic equations admitting $2^2 = 4$ different branches of solutions. After fixing this discrete choice at second order, the whole tower of higher-order coefficients will be completely and uniquely determined by just solving degree-one algebraic equations.

As required by the above argument, the particular choice of local $f$ having a global extremum at a given $\phi_0$ is relevant for discussing positive energy theorems and the non-perturbative stability of the related vacuum. Different choices of local branches turn out to define flat DW solutions interpolating between different vacua. The correct discrete choice for the SUSY extremum labeled by "2" proving the positive energy theorem turns out be[6]

$$ f^{(2)}(\phi_2) = \begin{pmatrix} -\frac{7\, 2^{2/3}}{\sqrt[4]{3}\, 5^{5/6}\ell} & -\frac{56}{5\sqrt[4]{3}\, 5^{5/6}\ell} \\ -\frac{56}{5\sqrt[4]{3}\, 5^{5/6}\ell} & \frac{2912\sqrt[3]{2}}{75\sqrt[4]{3}\, 5^{5/6}\ell} \end{pmatrix}, \tag{25} $$

the above values exactly matching the entries of the Hessian matrix of $f_{\text{SUSY}}$, evaluated at $\phi_2$. For the SO(7) and the G$_2$-invariant non-SUSY critical points, these are respectively given by

$$ f^{(2)}(\phi_1) = \begin{pmatrix} \frac{7(3+\sqrt{33})}{8\ell} & 0 \\ 0 & \frac{7(15+\sqrt{105})}{8\, 5^{2/3}\ell} \end{pmatrix}, \quad f^{(2)}(\phi_3) = \begin{pmatrix} \frac{7(3+\sqrt{33})}{\sqrt[3]{2}\, 3^{3/4} 5^{7/12}\ell} & 0 \\ 0 & \frac{28\sqrt[3]{2}(3+\sqrt{33})}{3\, 3^{3/4} 5^{7/12}\ell} \end{pmatrix}. \tag{26} $$

The perturbative analysis has been carried out up to order 15, where all equations are solved with impressively great accuracy in the region where all three critical points are located. We illustrate the results in Figure 2, where the scalar potential is plotted against the globally bounding functions $-3f^2$, for the three different choices of solutions for $f$, each of which satisfies all the hypotheses of the above positive energy theorem.

Further one-dimensional sections of the full plot are shown in Figures 3 and 4, where pairs of critical points can be studied together, and besides the existence of globally bounding functions for each of the critical points, we may find evidence for the existence of interpolating DW solutions.

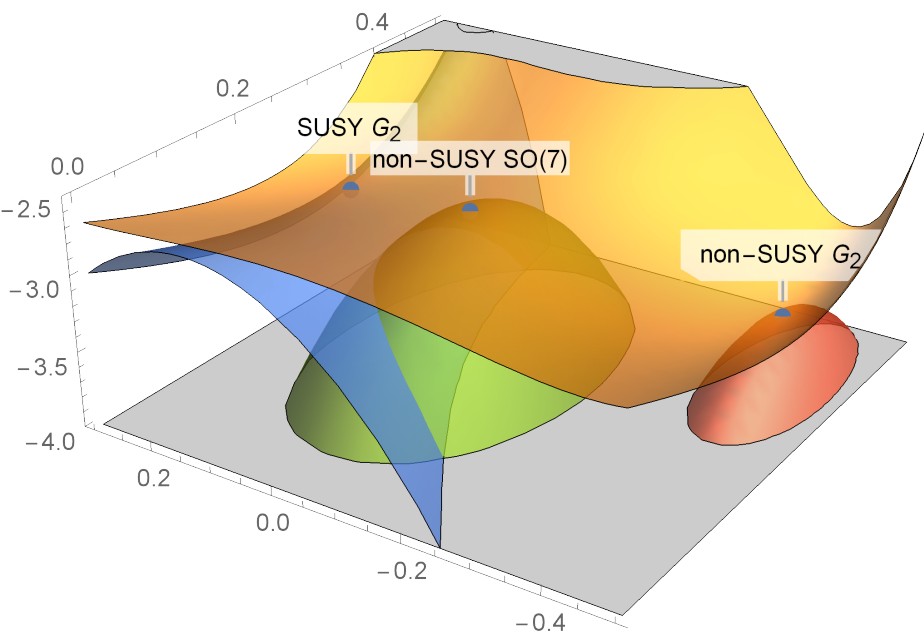

**Figure 2.** The profile of the scalar potential of $G_2$-invariant ISO(7) gauged supergravity (orange surface), against the three globally bounding functions $-3f^2$, obtained by solving the PDE (20) through a perturbative expansion around each critical point. The plot is drawn in $\ell = 1$ units.

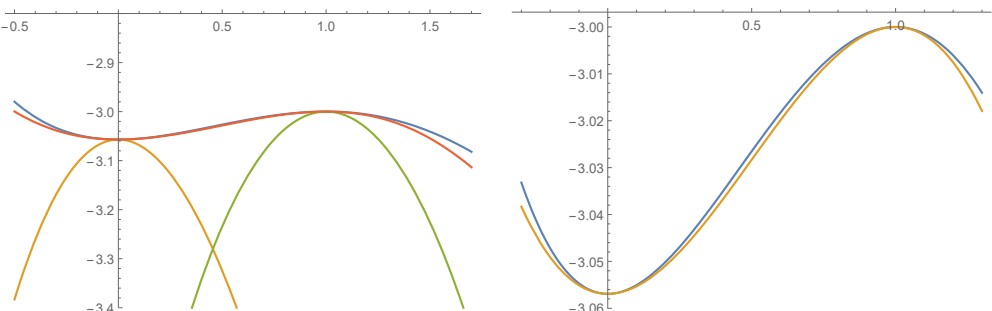

**Figure 3.** The profile of the scalar potential restricted to the straight line connecting vacua "1" and "2" (blue curve). Besides the previously determined globally bounding functions (in yellow and green), we also plot $-3f^2_{DW_{12}}$ (orange curve), having both critical points as local extrema, and thus defining the interpolating flow (zoomed on the right). The plots are both drawn in $\ell = 1$ units.

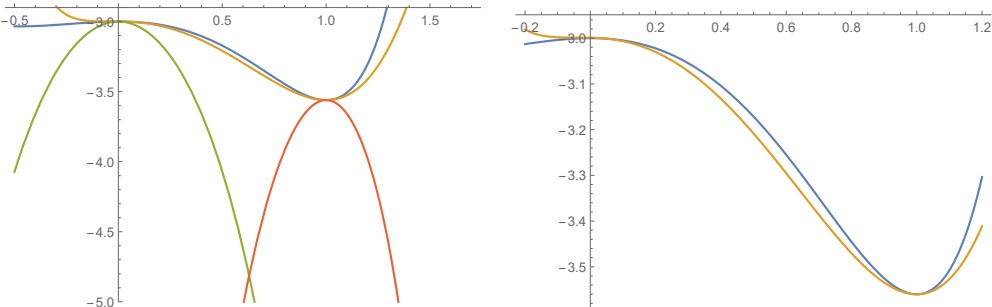

**Figure 4.** The profile of the scalar potential restricted to the straight line connecting vacua "1" and "3" (blue curve). Besides the previously determined globally bounding functions (in orange and green), we also plot $-3f^2_{DW_{13}}$ (yellow curve), having both critical points as local extrema, and thus defining the interpolating flow (zoomed on the right). The plots are both drawn in $\ell = 1$ units.

*Interpolating Flat DW Solutions*

As far as interpolating DWs are concerned, there turn out to exist solutions connecting the SO(7) vacuum with both G₂-preserving ones, while the only way of connecting these last two together seems to be the one going through the SO(7) saddle. The corrected discrete choices for the branches of second derivatives to find the DWs are

$$f^{(2)}(\phi_2) = \begin{pmatrix} \frac{7\sqrt[4]{3}(4\sqrt{6}-19)}{5\,10^{5/6}\ell} & -\frac{14\sqrt{2}}{5\,3^{3/4}5^{5/6}\ell} \\ -\frac{14\sqrt{2}}{5\,3^{3/4}5^{5/6}\ell} & \frac{56\sqrt[3]{2}(8\sqrt{3}-13\sqrt{2})}{25\,3^{3/4}5^{5/6}\ell} \end{pmatrix}, \quad f^{(2)}(\phi_3) = \begin{pmatrix} -\frac{7(\sqrt{33}-3)}{\sqrt[3]{2}\,3^{3/4}5^{7/12}\ell} & 0 \\ 0 & -\frac{28\sqrt[3]{2}(\sqrt{33}-3)}{3\,3^{3/4}5^{7/12}\ell} \end{pmatrix},$$

respectively, for the DW connecting "1" and "2" and the one connecting "1" and "3". The corresponding non-supersymmetric extremal flows were found by first determining the corresponding $f$'s up to order 15 in perturbation theory, and subsequently by numerically integrating the first-order HJ flow Equation (18), with initial conditions obtained by linearly perturbing the SO(7) vevs at large radial distance. The corresponding scalar radial profiles are plotted in Figures 5 and 6.

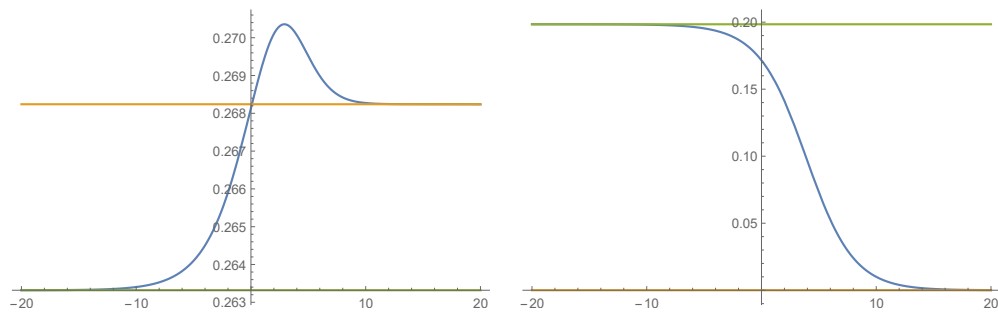

**Figure 5.** The profiles of $\varphi(r)$ (**left**) and $\chi(r)$ (**right**), plotted against $\frac{r}{\ell}$. The scalar vevs asymptote to the ones in the SUSY-G₂ vacuum as $r \to -\infty$, while they approach those in the non-SUSY SO(7) as $r \to +\infty$.

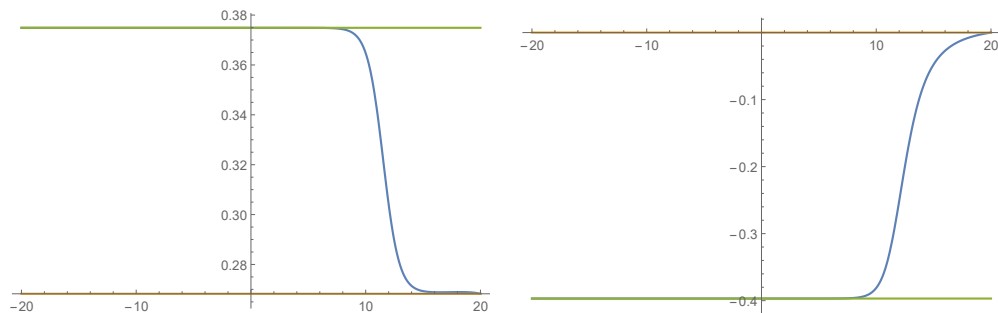

**Figure 6.** The profiles of $\varphi(r)$ (**left**) and $\chi(r)$ (**right**), plotted against $\frac{r}{\ell}$. The scalar vevs asymptote to the ones in the non-SUSY-G₂ vacuum as $r \to -\infty$, while they approach those in the non-SUSY SO(7) as $r \to +\infty$.

It may be worth mentioning that such profiles need not be *monotic*. What has to satisfy monotonicity is the function $f$ when projected along the flow, and indeed, in both situations, this turns out to be the case.

## 6. Curved DW Geometries: A Non-Separable HJ Treatment

Let us now conclude by making some comments concerning the case of a curved DW. While the study of flat flows may be (besides being easier) useful in order to exclude non-perturbative decay, when looking for explicit bubble solutions, it is dS curved DWs what we should look for. Indeed, with a suitable choice of boundary conditions, when the 3D slices of our 4D geometry are positively curved, they may describe expanding bubbles

and non-perturbative instabilities of different sorts, as highlighted in Section 3. The relevant 4D *ansatz* now becomes

$$ds_4^2 = e^{2A} L^2 ds_{\text{dS}_3}^2 + e^{2B} dr^2, \tag{27}$$

where we explicitly pulled out the dS radius $L$ in such a way that the exponential warp factors appearing in the above metric are dimensionless.

The whole Hamiltonian formulation goes through almost identically to the previous flat case studied in Section 4, and the resulting Hamiltonian precisely coincides with the one in (17), except that the effective potential now has an extra term induced by the dS curvature, namely[7]

$$V_{\text{eff}}^{(\text{curved})} = V_{\text{eff}}^{(\text{flat})} - \frac{3}{L^2} e^A. \tag{28}$$

It may be worth commenting that, even though it seems a very marginal complication, it is in fact rather a drastic one. The reason for this is that adding such a potential term spoils the factorized structure of the $A$ dependence, which allowed us to decouple $A$ from the system, when studying the flat case. Thus, we will now have to face a genuine three-field situation.

We propose here an appropriate generalization of the perturbative procedure introduced in [23] and applied here in Section 4, which in principle allows one to determine $F(A, \phi^i)$ at arbitrarily high orders. Due to the non-separability of our effective potential, we have to cast a more general *ansatz* for $F$:

$$F = \sum_{k=0}^{\infty} F^{(k)}(A) \frac{(\phi - \phi_0)^k}{k!}, \tag{29}$$

where the coefficients of our perturbative expansion are now promoted to functions of $A$.

The HJ constraint (19) specified for the *ansatz* (29) will now impose a set of ODEs (each for any perturbative order) for the functions $F^{(k)}(A)$ rather than algebraic conditions, as it was in the separable case. Nevertheless, the condition $F^{(1)}(A) = 0$ still makes sure that the whole perturbative tower remains decoupled. At zeroth-order, one finds:

$$\frac{1}{3} e^{-3A} (\dot{F}^{(0)})^2 - \frac{3}{\ell^2} e^{3A} - \frac{3}{L^2} e^A = 0, \tag{30}$$

where the dot denotes differentiation with respect to $A$, and we set $V(\phi_0) \equiv -\frac{3}{\ell^2}$. The above ODE is then easily solved by[8]

$$F^{(0)} = \frac{e^{3A}}{\ell} \left( 1 + \frac{\ell^2}{L^2} e^{-2A} \right)^{3/2}. \tag{31}$$

Subsequently, in analogy with the flat case, the only crucial step in the whole algorithm is determining $F^{(2)}$, as it satisfies a *non-linear* ODE. Generically, this will admit a continuous family of solutions depending on the choice of the integration constants $c_{ij}$ (each for every independent component of $F^{(2)}$). Once that one is fixed, higher perturbative orders will give rise to *linear* ODEs fixing each $F^{(k)}(A)$ in terms of the previously determined ones.

For our specific case,

$$F^{(2)} = \begin{pmatrix} F_{\varphi\varphi}^{(2)} & F_{\varphi\chi}^{(2)} \\ F_{\chi\varphi}^{(2)} & F_{\chi\chi}^{(2)} \end{pmatrix}, \tag{32}$$

whose entries are fixed by the following system of first-order ODEs:

$$\begin{cases} \frac{2}{3} \dot{F}^{(0)} \dot{F}_{\varphi\varphi}^{(2)} - \frac{8}{7} (F_{\varphi\varphi}^{(2)})^2 - \frac{8}{7} e^{-2\varphi_0} (F_{\varphi\chi}^{(2)})^2 + e^{6A} V_{\varphi\varphi}(\varphi_0, \chi_0) = 0, \\ \frac{2}{3} \dot{F}^{(0)} \dot{F}_{\chi\chi}^{(2)} - \frac{8}{7} (F_{\varphi\chi}^{(2)})^2 - \frac{8}{7} e^{-2\varphi_0} (F_{\chi\chi}^{(2)})^2 + e^{6A} V_{\chi\chi}(\varphi_0, \chi_0) = 0, \\ \frac{2}{3} \dot{F}^{(0)} \dot{F}_{\varphi\chi}^{(2)} - \frac{8}{7} F_{\varphi\varphi}^{(2)} F_{\varphi\chi}^{(2)} - \frac{8}{7} e^{-2\varphi_0} F_{\varphi\chi}^{(2)} F_{\chi\chi}^{(2)} + e^{6A} V_{\chi\varphi}(\varphi_0, \chi_0) = 0, \end{cases} \tag{33}$$

where $V_{ij}(\varphi_0, \chi_0)$ and $\varphi_0$ are known data concerning the AdS vacuum and the function $F^{(0)}(A)$ was determined in (31) by solving the HJ constraint at zeroth order. After choosing a particular solution to the above first-order system, all higher-order coefficients $F^{(k)}(A)$ for $k > 2$ will be fixed by solving *linear* ODEs. This concrete procedure offers a viable numerical method to integrate the HJ PDE defining $F$, even in the case of curved flows, i.e., where the separable *ansatz* no longer works.

## 7. Concluding Remarks

In this paper, we discussed positive energy theorems for non-SUSY AdS extrema within the $G_2$-invariant sector of maximal supergravity with the ISO(7) gauge group. Given the massive IIA origin of the theory, this analysis is directly relevant to the issue of the non-perturbative stability of non-SUSY AdS. Our results show the existence of a dynamical protection mechanism against non-perturbative decay of the non-SUSY vacua, enforced by the validity of a positive energy theorem. We were able to prove this by using HJ techniques.

However, it is worth remarking that this result by no means guarantees full non-perturbative stability of the non-SUSY vacua at hand. If nothing else, this is because the $\mathcal{N} = 1$ model we adopted is *not* a good effective description in a physical sense, since it does not capture the lightest modes. Of course, it is still a valid description in a mathematical sense, meaning that it describes a consistent set of 10D deformations of the metric and the rest of the IIA fields. In this context then, a fair conclusion to our analysis is the statement that possible non-perturbative instabilities of the non-SUSY $G_2$ vacuum have to be searched for within a different setup, because the positive energy theorem proven in this paper prevents the existence of any expanding bubble solutions within this truncation.

Thinking along the lines of [8], we actually know that, if an instability exists, it will be of a brane nucleation type, due to the presence of a Freund–Rubin flux $F_{(6)}$ that fills internal space completely. The source for this flux singularity has to be a D2 brane. The corresponding metric and dilaton would then be of the form

$$
\begin{aligned}
ds_{10}^2 &= H_{\text{D2}}^{-1/2}\, ds_3^2 + H_{\text{D2}}^{1/2} \left( dr^2 + r^2\, ds_{S^6}^2 \right), \\
e^\Phi &= H_{\text{D2}}^{1/4},
\end{aligned}
\tag{34}
$$

where $H_{\text{D2}}$ is a function of $r$. Now, for a fully localized D2 within the transverse $\mathbb{R}^7$, this function must be given by $(1 + \frac{Q_{\text{D2}}}{r^5})$, which behaves as $r^{-5}$ close to the source. It is straightforward to check that this behavior respects the truncation *ansatz* in (A1). In particular, a near-horizon D2 solution is given by the following 4D scalar configuration:

$$
e^\varphi \sim r^{-1/2}, \qquad \chi = \text{const.}
\tag{35}
$$

As a consequence, any brane nucleation process involving a fully localized D2 should be captured by our minimal supergravity model. Because we have just proven that such instabilities cannot occur in our model, we have to conclude that an instability of this vacuum must necessarily involve non-spherically symmetric D2 charge distributions.

Indeed, in [13], the non-perturbative instability that the non-SUSY vacua are found to suffer from corresponds to D2 brane charge, which is *smeared* over $S^6$. Note that this requires scaling the internal components of the metric along $S^6$ and the 10D dilaton independently, and this goes manifestly beyond the truncation *ansatz* in (A1).

**Funding:** This research received no external funding.

**Institutional Review Board Statement:** Not applicable.

**Informed Consent Statement:** Not applicable.

**Data Availability Statement:** Not applicable.

**Acknowledgments:** We would like to thank Pieter Bomans, Davide Cassani, and Nicolò Petri for collaborating in a related project. We further thank Nicolò Petri for some valuable comments on a draft version of this manuscript. The work of G.D. is supported by the STARS at unipd grant named THEsPIAN.

**Conflicts of Interest:** The author declares no conflict of interest.

## Appendix A. Consistent Truncations of Massive Type IIA Supergravity on $S^6$

In this Appendix, we include some details concerning the 10D origin of $G_2$-invariant ISO(7) gauged supergravity in four dimensions. Massive type IIA supergravity admits a consistent truncation on the six-sphere, for which the full non-linear KK *ansatz* was determined in [19]. When restricting to the $G_2$-invariant sector thereof, this reduction precisely turns out to yield the minimal supergravity theory discussed in Section 2. Its field content is simply given by the 4D metric $g_{\mu\nu}$ and the complex scalar field $S = \chi + ie^{-\varphi}$.

The full reduction *Ansatz* for the 10D fields in the *string frame* reads

$$ds_{10}^2 = e^{2\varphi} ds_4^2 + g^{-2} e^{\varphi} \left(1 + e^{2\varphi}\chi^2\right)^{-1} ds_{S^6}^2 \,,$$

$$e^{\Phi} = e^{5\varphi/2} \left(1 + e^{2\varphi}\chi^2\right)^{-3/2} \,,$$

$$B_{(2)} = g^{-2} e^{2\varphi} \chi \left(1 + e^{2\varphi}\chi^2\right)^{-1} J \,, \tag{A1}$$

$$C_{(1)} = 0 \,,$$

$$C_{(3)} = \tilde{C}_{(3)} + g^{-3} \chi \, \Omega_I \,,$$

where $ds_4^2$ denotes the 4D metric, $ds_{S^6}^2$ the unit six-sphere metric, while $J$ and $\Omega$ define the nearly Kähler (NK) structure of $S^6$, and $\tilde{C}_{(3)}$ represents a 4D three-form field satisfying

$$d\tilde{C}_{(3)} \stackrel{!}{=} \left( g \, e^{\varphi} \left(1 + e^{2\varphi}\chi^2\right)^2 \left(5 - 7e^{2\varphi}\chi^2\right) + m \, e^{7\varphi}\chi^3 \right) \text{vol}_4. \tag{A2}$$

The above expressions also contain the constant $g$, which appears in the effective scalar potential (12) together with Romans' mass $m$ and is identified with the gauge coupling.

The consistency of the truncation automatically guarantees the equivalence between the 4D equations of motion obtained from the reduced Lagrangian (9) of ISO(7) gauged supergravity and the 10D equations of motion of massive IIA supergravity. In particular, by virtue of the uplift formulae given in (A1), the three-vacuum solution collected in Table 1 may be directly interpreted as AdS$_4 \times S^6$ solutions of massive IIA supergravity. More specifically, the SO(7)-preserving vacuum is simply of a Freund–Rubin type, i.e., where the only non-vanishing flux (besides of course $F_{(0)} = m$) is the (magnetic) six-form flux filling internal space completely. In the $G_2$-invariant vacua, on the other hand, the $S^6$ retains its round metric, while non-trivial lower-form internal fluxes such as $F_{(2)}$, $F_{(4)}$, and $H_{(3)}$ are turned on by exploiting the fundamental two- and three-forms defined by its NK structure.

## Notes

1. From now on, we set $\kappa_4 = 1$.
2. From now on, we will fix $g = m$.
3. It is worth mentioning that, to obtain this reduced 1D Lagrangian, an integration by parts is required in order to eliminate the term with the second derivative $A''$.
4. We may set $B = 0$ here, since it is a completely irrelevant overall factor. However, as pointed out earlier, the first-order flow equations are sensitive to the choice of $B$ and may simplify significantly in particularly clever gauge choices.

5   In this general formalism, we collectively denote the scalar fields by $\phi \equiv (\varphi, \chi)$. As a consequence, the $k$th derivative of a scalar function $f$ is represented by a rank $k$ tensor field.

6   We have fixed $m = \frac{2}{5^{7/12}} \ell^{-1}$ in such a way that the value of the cc in the SO(7) vacuum equals $-\frac{3}{\ell^2}$.

7   We still stick to the $B = 0$ gauge for the sake of simplicity here.

8   Note that it correctly reproduces the flat result $F^{(0)} = \frac{e^{3A}}{\ell}$ in the limit where $L \gg \ell$.

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
