# Peer review of "Positive Energy and Non-SUSY Flows in ISO(7) Gauged Supergravity"

_universe, doi:10.3390/universe8050293_

Round 1
Reviewer 1 Report
This paper studied the positive energy theorem in a non-supersymmetric AdS extrema. By using the Hamiltonian-Jocobi formalism, the author can perturb this theory in a first order fomulation. This process helps the author to study the positive energy theorem of the non-supersymmetric extrema. From my point of view, this is a reasonable treatment and no doubt that it would provide some new things for the interested readers.
Moreover, the author continued studying the domain walls between these extrema, and perturbative Hamiltonian-Jacobi technique was discussed to study this topological domain walls.
I found this paper was interesting to this field, and the organizaiton and langurage were also suitable. Therefore, I recommend to accept this paper.
Author Response
I thank the referee for all of her/his comments.
Reviewer 2 Report
This is an interesting paper, addressing the non-perturbative stability of a recently-discovered perturbatively stable non-supersymmetric AdS_4 vacuum. In particular, the author constructs a positive mass theorem within the G_2 invariant sector of ISO(7) gauged supergravity. I would like the following issues to be improved/addressed before publication:
1) My main criticism with the paper is the somewhat poor presentation. In particular, there are various typos, e.g. already on the first page "estimatetd", "dyanmics", and so it continues throughout the paper.
2) Moreover, the presentation of non-perturbative instabilities in section 3 is not very clear and should be expanded so that less expert readers can understand the paper. This applies most prominently to the list of criteria in lines 163 - 167.
3) The word "effective" is used repeatedly throughout (e.g. lines 159, 216, 242, 303, 344), but it is very unclear what is meant by it. Indeed, as the author says, the ISO(7) theory, and the G_2 invariant sector are not effective theories in the Wilsonian sense. So why does the author repeatedly refer to these various theories as "effective"? What does that word mean? Presumably, the author wants to refer that they are working within a consistent truncation. This should be explained appropriately. Indeed, this point is crucial: if the subsector studied were an effective theory, then the lack of instabilities within this subsector would be sufficient to show that there is no instability in the entire theory.
4) Why is it consistent to simply plug in the Ansatz (4.1) into the action (2.9)? Normally, such an Ansatz should be plugged into the equations of motion, unless symmetry arguments or otherwise imply that the Ansatz (4.1) is a consistent truncation of the theory. If this is not the case, then reduction of (2.9) by the Ansatz (4.1) may yield solutions that do not solve the full equations of motion.
5) What is F in equation (4.5)? This should be explained better.
Once these points are addressed, I am happy to recommend publication.
Round 2
Reviewer 2 Report
I am happy with all but one of the changes made by the author. The word "effective" should not be used to describe the consistent truncation procedure described by the author. The author themselves agree that this is a misleading terminology. The actions obtained by the reduction could instead be referred to as reduced actions/theories or truncated actions/theories. With this change, I am happy to recommend publication.
Author Response
Dear Editor,
I took care of last minor corrections requested by the referees. Minor changes are in bold.